# Context-Aware Alignment: Adapting Large Language Models to Individual Historical Data

## Abstract

Aligning large language models with human preferences is essential for ensuring their effectiveness, utility, and safety in real-world applications. While much of the current research focuses on aligning LLMs with generalized human values such as fairness, transparency, and ethical behavior, limited attention has been given to aligning LLMs with the preferences and characteristics of individual users. In this paper, we propose a novel approach that leverages individual historical context to achieve personalized alignment, adapting LLMs to align with the unique traits and preferences of specific users. Our method focuses on extracting persona-related representations—abstract features encapsulating conversational style, tone, and preferences—from past user interactions. These representations guide the model in generating responses tailored to the user's individual characteristics. Experimental results demonstrate that our approach significantly outperforms existing baselines, improving the model's ability to reflect individual personas while maintaining contextual appropriateness. This research opens new possibilities for more personalized, context-aware, and user-centric applications of LLMs.

## 1 Introduction

Large language models (LLMs) have shown tremendous potential in a wide range of tasks, but their effectiveness is highly contingent on their ability to respond in ways that align with human values (Brown et al., 2020; Bubeck et al., 2023; Touvron et al., 2023). Aligning LLMs with human preferences is essential for ensuring their utility, safety, and overall effectiveness in real-world applications (Liu et al., 2024; Ouyang et al., 2022b; Rafailov et al., 2023b). Most existing research in alignment has focused on generalizing models to reflect broad, high-level human values such as fairness, transparency, and ethical behavior. While these efforts are critical, they fail to address the needs of specific individuals, which limits the potential of LLMs in personalized contexts. In applications like personalized AI assistants, user-specific interaction models are necessary for creating adaptive, engaging, and meaningful interactions (Li et al., 2024; Kirk et al., 2024; Shi et al., 2024).

Achieving personalized alignment is a significant challenge, as it requires LLMs to not only reflect generalized values but also adapt to the specific conversational style, tone, and needs of each individual user. Traditional alignment methods, such as Direct Preference Optimization (DPO), focus on aligning models to general preferences using feedback on preferred and non-preferred outputs (Wu et al., 2024; Rafailov et al., 2023a). While DPO has shown promise in some scenarios, it has limitations when applied to individual alignment. These methods typically rely on binary feedback, treating general human values as universal preferences. However, this can lead to issues where what is considered a general preference for human alignment may not align with the specific traits or style of a given individual. When such a heterogeneous mix of preference data is used for training, the model receives conflicting optimization signals. This can lead to a state of 'policy confusion,' where the model struggles to learn a coherent strategy that effectively balances the nuanced requirements of individual alignment with the broader constraints of general alignment. The model's inability to distinguish between universal values and individual preferences can lead to suboptimal personalization, undermining the adaptability needed for real-world applications.

To address these limitations, we propose a novel approach that explicitly leverages individual historical context to align LLMs with user-specific preferences. Our method begins by extracting persona-related representations from past user interactions. These representations encapsulate abstract features such as conversational tendencies, tone, and preferences, forming a comprehensive profile of the user's unique characteristics. We employ a contrastive learning framework to derive these persona embeddings, ensuring that they are both robust and representative of the user's historical context.

Building on this foundation, we introduce a Representation Regularization Loss (**RE**) within the DPO framework, referred to as **DPORE**, to guide the alignment process. This loss minimizes the divergence between the model's outputs and the user's persona representations, enabling the model to consistently generate responses that reflect individual traits. By integrating these user-specific embeddings into the training process, we ensure that the model's outputs are both personalized and contextually appropriate.

Our approach was evaluated through comprehensive experiments against established baselines, including traditional DPO methods. Results demonstrate that DPORE significantly enhances the model's ability to align with individual personas, as measured by both automated metrics and human evaluations. Notably, this improvement in personalization does not come at the expense of the model's general-purpose performance, confirming the efficacy of our regularization technique in striking a balance between specific user needs and broad utility. Instead, it highlights the potential of leveraging individual historical context as a transformative step in alignment research, offering a principled way to achieve deeper personalization.

## 2 RELATED WORK

### 2.1 HUMAN ALIGNMENT OF LLMS

Aligning LLMs with human preferences is essential for ensuring that their outputs are both useful and ethical in real-world applications. Early alignment methods primarily focused on aligning LLMs with generalized human values, such as fairness, safety, and transparency. Among these, Reinforcement Learning from Human Feedback (RLHF) has emerged as one of the most successful techniques (Ouyang et al., 2022a). RLHF uses human feedback to fine-tune LLMs, improving their alignment with human preferences.

To address the computational cost and stability challenges associated with RLHF, Direct Preference Optimization (DPO) (Rafailov et al., 2023a) was proposed as a more efficient alternative. DPO optimizes language models by increasing the relative probability of preferred responses over dispreferred ones, reducing reliance on complex human feedback loops. Another promising direction in alignment research is representation alignment (Liu et al., 2024), which focuses on identifying representations of high-level human preferences embedded within LLMs. By modifying these representations, researchers can achieve more precise control over model behavior.

However, while these approaches excel at aligning models with generalized human values, they often overlook the personalized needs of individual users. Existing methods are typically designed to align models with societal-level values, leaving the nuanced preferences and unique characteristics of specific individuals underexplored.

### 2.2 INDIVIDUAL ALIGNMENT

The challenge of aligning LLMs with individual user preferences has garnered growing attention. Some research has explored customized LLMs that leverage user-specific information, such as interaction history, to adapt model responses. For instance, USER-LLM (Ning et al., 2024) integrates user embeddings to contextualize LLMs based on interaction history, enabling more personalized outputs. However, this approach requires a rich history of user interactions, limiting its effectiveness for new or infrequent users lacking sufficient data.

ALOE (Wu et al., 2024) adopts a different strategy by implicitly inferring user preferences from multi-turn dialogues. It dynamically adjusts the model's behavior based on inferred preferences and employs DPO for training. However, like other DPO-based approaches, ALOE struggles to directly capture personalized features, as DPO optimization often emphasizes overall response quality. This

focus can lead to distortion of learned preferences by irrelevant factors, such as tone or response length.

In contrast, our approach explicitly extracts user-specific features from input data and aligns the model with these features during training. This direct alignment ensures that the model learns user-specific characteristics without being influenced by irrelevant factors, enabling more accurate personalization.

## 2.3 PERSONALIZED ALIGNMENT

A related area of research is personalized alignment (Tu et al., 2023; Li et al., 2023a; Chen et al., 2024), which aims to enhance specific personality traits or behaviors in LLMs, such as humor, courage, or politeness. These methods focus on making models more engaging by training them to express predefined traits.

While personalized alignment typically involves tailoring the model to exhibit broad personality traits, our approach dynamically adapts to individual interactions. This flexibility ensures that the model's responses not only align with general personality traits but also reflect the specific preferences and conversational nuances of each user. By bridging the gap between generalized personality adaptation and individual alignment, our method enables a more comprehensive and user-centric approach to personalization.

## 3 METHODS

Large language models have demonstrated impressive capabilities in generating human-like text. However, aligning their responses with individual user preferences remains a significant challenge. In this section, we detail our method for achieving personalized alignment, starting with dataset preparation and progressing through embedding learning, fine-tuning, and the incorporation of a novel representation regularization technique. Figure 1 illustrates an overview of the training and deployment pipeline for our approach.

## 3.1 DATASETS

The dataset used in this study is derived from the ALOE benchmark introduced by Wu et al. (2024). It comprises over $3,000$ multi-turn interaction samples, created using a persona pool of $3,310$ unique user personas. Each sample includes both preferred responses, which are tailored to specific user personas, and non-preferred responses, which are generic and disregard persona information. To construct the persona pool, an iterative self-generation and filtering process was employed, ensuring diverse and distinct user profiles and personalities. This approach provides a rich dataset for studying dynamic, personalized alignment in conversational settings. The multi-turn nature of the dataset ensures that the alignment is evaluated not only on a single response but across the progression of a conversation, capturing more realistic interaction scenarios.

## 3.2 USER EMBEDDING LEARNING

ALOE aligns large language models with individual preferences relying on DPO, focus on learning from positive and negative response samples. However, this can cause issues when general preferences, which may work for societal-level alignment, conflict with the specific traits or conversational style of an individual user. When such mixed samples are used in training, the model becomes confused and struggles to optimize both individual and general alignment simultaneously.

To address these challenges and to create robust, explicit user persona representations, we propose a multi-view learning approach to extract user-specific persona representations from multi-turn interactions. Drawing inspiration from multi-view learning techniques, which have proven highly effective in domains like image recognition and natural language understanding for learning invariant features, we focus on isolating shared persona-related features across different dialogue excerpts while ignoring extraneous information or topic-specific noise. This allows our model to learn a condensed, meaningful representation of who the user is, rather than what they are talking about.

As illustrated in Figure 1, this method begins by sampling two dialogue excerpts, denoted as $x_1$ and $x_2$, that are associated with the same user persona $p$. These two excerpts form a **positive**

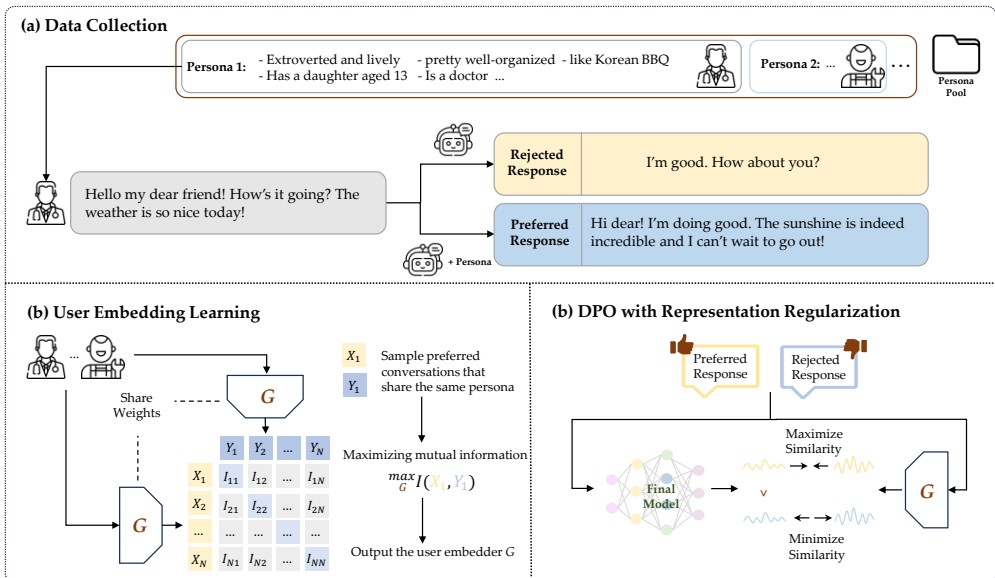

Figure 1: Overview of the Proposed Method for Individual Alignment. **(a) Data Collection:** Multi-turn dialogue samples are generated using a persona pool, with each sample containing a preferred and a rejected response. **(b) User Embedding Learning:** A contrastive learning framework is used to train a user embedding extractor $G$, which learns persona-specific features by maximizing mutual information between conversations sharing the same persona. **(c) DPO with Representation Regularization:** The DPO framework is extended with a representation regularization term that aligns model outputs with user embeddings for preferred responses and penalizes alignment for rejected responses.

**pair**, as they originate from the same underlying user identity. Conversely, dialogue excerpts from different personas are treated as **negative pairs**. We then employ a contrastive learning framework to maximize the similarity between positive pairs and minimize the similarity between negative pairs. This encourages the feature extractor, denoted as $G$, to learn compressed representations that robustly capture shared persona-related traits while actively discarding irrelevant noise, conversational artifacts, or topic-specific details.

Mathematically, given a set of dialogue excerpts $\{x_i\}$ and their corresponding personas $\{p_i\}$, we aim to train an encoder $G$ to map each dialogue excerpt $x_i$ into a low-dimensional embedding $G(x_i)$. The objective is to ensure that embeddings from the same persona are close, and embeddings from different personas are far apart. We use a **contrastive loss function**, specifically a variant of the InfoNCE loss, for this purpose. For a given anchor dialogue excerpt $x_a$ associated with persona $p_a$, we consider a positive sample $x_p$ (another dialogue excerpt from the same persona $p_a$) and a set of negative samples $\{x_n^k\}_{k=1}^K$ (dialogue excerpts from different personas). The contrastive loss for this anchor is defined as:

$$L_{\text{contrastive}}(x_a, x_p, \{x_n^k\}_{k=1}^K) = -\log \frac{\exp(\text{sim}(G(x_a), G(x_p))/\tau)}{\exp(\text{sim}(G(x_a), G(x_p))/\tau) + \sum_{k=1}^K \exp(\text{sim}(G(x_a), G(x_n^k))/\tau)}$$

(1)

where $\text{sim}(\cdot, \cdot)$ is a similarity function (e.g., cosine similarity), and $\tau$ is a temperature hyperparameter that controls the sharpness of the distribution. By minimizing this loss over numerous anchor-positive-negative triplets, the encoder $G$ learns to produce robust user persona embeddings. For each user, we utilized Llama3-8B-Instruct to generate 10 distinct dialogues, providing sufficient data for learning these persona embeddings.

To further enhance robustness and prevent overfitting to superficial textual features, we introduce random truncation of dialogues during training. This technique reduces the influence of dialogue length and specific structural elements on the learned embeddings, forcing the model to focus on intrinsic persona-related characteristics. The resulting feature extractor $G$ encodes the unique persona traits of users based on their historical prompts and conversational context. These highly distilled

| Method | Turn | | | | | | | | | | Avg. |
|--------|------|------|------|------|------|------|------|------|------|------|------|
| | $k=1$ | $k=2$ | $k=3$ | $k=4$ | $k=5$ | $k=6$ | $k=7$ | $k=8$ | $k=9$ | $k=10$ | |
| **Base Model** | 3.692 | 3.457 | 3.660 | 3.540 | 3.492 | 3.456 | 3.421 | 3.448 | 3.397 | 3.415 | 3.497 |
| **Preferred-SFT** | 4.264 | 4.006 | 3.990 | 4.090 | 4.108 | 4.070 | 4.134 | 4.116 | 4.128 | 4.182 | 4.108 |
| **DPO** | 4.356 | **4.514** | 4.426 | 4.436 | 4.442 | 4.456 | 4.426 | 4.446 | 4.442 | 4.420 | 4.436 |
| **DPORE** | **4.583** | 4.448 | **4.498** | **4.544** | **4.580** | **4.588** | **4.542** | **4.588** | **4.550** | **4.562** | **4.543** |

Table 1: Alignment Levels Across k-th Turn in Multi-Turn Dialogues for Various Models and Methods. This table compares the performance of different alignment methods: Base, SFT, DPO, and our proposed method on Llama3-8B-Instruct. The columns represent the alignment scores at each turn ($k = 1$ to $k = 10$) in a multi-turn dialogue, while the final column reports the average score across all turns. Higher scores indicate better alignment with individual user personas. The results show that our method consistently achieves the highest alignment scores, outperforming the baselines and demonstrating its effectiveness in adapting to user-specific preferences over extended interactions.

persona embeddings are then incorporated into the preference alignment training process, enabling the main LLM to generate responses that precisely reflect user-specific styles, tones, and preferences, leading to a much more granular and faithful personalization.

### 3.3 DPO with Representation Regularization

To further refine the alignment process, we extend the DPO framework with a novel representation regularization term. This regularization leverages the user embeddings, $G(m_i, p_i)$, to guide the model towards generating responses that are aligned with individual user preferences. This explicit guidance helps resolve the policy confusion that often arises when general and individual preferences are mixed.

For a given prompt $m_i$, positive response $p_i$, and negative response $r_i$, we align the model's output, $y_\theta(m_i, p_i)$, with the persona embedding $G(m_i, p_i)$ using a projection head $h_\phi$. The regularization loss is defined as:

$$L_{\text{reg}} = \lambda \cdot \text{sim}\big(y_\theta(m_i, p_i), h_\phi(G(m_i, p_i))\big) - \lambda \cdot \text{sim}\big(y_\theta(m_i, r_i), h_\phi(G(m_i, r_i))\big),$$

where $\text{sim}(\cdot, \cdot)$ is a similarity metric (e.g., cosine similarity) and $\lambda$ controls the balance between preference optimization and regularization. The regularization term is designed to enforce that the preferred response is close to the persona embedding, while non-preferred responses are penalized for deviating from it. This alignment ensures that the model produces outputs that are consistent with the specific user's persona. The total loss combines the DPO objective and the regularization term:

$$L_{\text{total}} = \sum_{i=1}^{K} \log \sigma \Bigg( \beta \cdot (\log \frac{y_\theta(p_i \mid m_i)}{y'_\theta(p_i \mid m_i)} - \log \frac{y_\theta(r_i \mid m_i)}{y'_\theta(r_i \mid m_i)})$$

$$+ \lambda \cdot (\text{sim}\big(y_\theta(m_i, p_i), h_\phi(G(m_i, p_i))\big) - \text{sim}\big(y_\theta(m_i, r_i), h_\phi(G(m_i, r_i))\big)) \Bigg),$$

where $y_\theta$ and $y'_\theta$ denote the trained and reference models, respectively. The additional regularization ensures alignment with user persona embeddings for preferred responses and penalizes alignment for rejected responses. This mechanism reduces the influence of irrelevant factors and enhances personalization.

By integrating user representation regularization into DPO, our method improves the model's ability to generate responses that are both contextually appropriate and tailored to individual user preferences. This approach bridges the gap between general alignment and personalized interaction, offering a scalable solution for dynamic, user-specific alignment.

# 4 EXPERIMENTS

To validate the effectiveness of our method, we follow Wu et al. (2024) to conduct experiments on multi-turn dialogue tasks. Here, we compared various baseline approaches using both human evaluations and automated assessments. Given the trade-offs between enhancing specific capabilities and maintaining overall performance, we further assess the nominal performance of different methods. We also performed hyper-parameter experiments to assess the impact of representation regularization weights on performance. Finally, we present visualizations to intuitively demonstrate the effectiveness of our approach. Finally, we present qualitative visualizations through a case study to intuitively demonstrate the effectiveness and nuanced improvements of our approach in real-world conversational settings.

## 4.1 EXPERIMENTAL SETUPS

**Dataset**    The dataset used for training and testing is derived from the ALOE benchmark (Wu et al., 2024). The dataset consists of over 3,000 multi-turn dialogue samples generated using a persona pool of 3,310 unique user personas. Each sample contains ten rounds of dialogue, with both preferred responses (tailored to user personas) and non-preferred responses (generic, persona-agnostic). For evaluation, we use a subset of 100 evaluation cases, each associated with a distinct user persona, including detailed profile and personality descriptions. These evaluation cases are used to conduct role-playing experiments where GPT-based agents interact with our trained models.

**Baselines**    To evaluate the efficacy of our proposed approach, we conducted extensive comparisons with existing methods, which includes:

- **Base Model** This baseline consists of the selected instruction-tuned LLMs that have been aligned with general human preferences but lack explicit mechanisms for individual alignment. It serves to highlight the limitations of general alignment in capturing nuanced, user-specific preferences. In this study, we selected Llama-3-8B-Instruct (AI@Meta, 2024) as the base model.
- **Preferred-SFT** This baseline involves fine-tuning the language models using only the preferred responses from the dataset. To ensure nominal performance is maintained, we follow prior work and mix additional general-purpose tasks from the UltraFeedback dataset (Cui et al., 2023). Each example in UltraFeedback contains human preferences for general query responses, and we use only the preferred responses for SFT training.
- **DPO** Direct Preference Optimization (Rafailov et al., 2023a) optimizes LLMs without explicit user feature extraction. Like Preferred-SFT, DPO incorporates data from the UltraFeedback dataset to improve nominal performance. In our proposed method, representation regularization is applied only to individual preference data, not to general preference data, ensuring precise personalization.

Further implementation details for the baselines and our approach are provided in Appendix B.

## 4.2 RESULTS

To evaluate the effectiveness of our method in aligning models with individual user preferences, we conducted automated evaluations comparing our approach with several baseline methodologies. The evaluation primarily assessed how well the generated responses aligned with user personas and profiles. Leveraging GPT-4o mini for evaluation, which correlates strongly with human judgments (Li et al., 2023b), we provided prompts containing model-generated responses alongside agent persona descriptions. The consistent superiority of DPORE, as shown in Table 1, is particularly significant. While other methods may perform reasonably well in early turns, DPORE maintains a high level of alignment even in later stages of the conversation (k=8, 9, 10). This suggests that the persona representations learned by our method provide a stable and continuous signal throughout the interaction, preventing the model from reverting to a generic, non-personalized mode, which is a common pitfall in long-form dialogue generation. The prompts used are detailed in Appendix A.

To mitigate the variance inherent in GPT-based evaluations, we set the generation temperature to 0 and conducted five evaluation runs for each method. The results, summarized in Table 1, demonstrate that our method consistently outperformed baseline approaches across nearly all dialogue rounds and average scores. This performance improvement stems from our explicit extraction of user-specific

features, which ensures that contrastive learning focuses on persona-related differences between positive and negative samples rather than irrelevant factors such as response length.

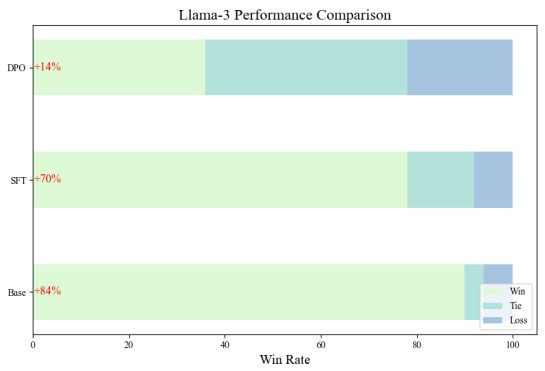

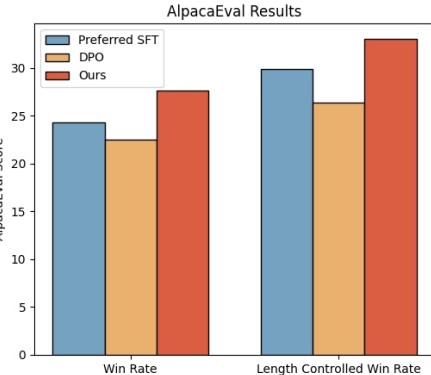

(a) Human evaluation on individual alignment.

(b) AlpacaEval win rate.

Figure 2: Model performance evaluation. (a) Human evaluation results comparing our individual-aligned LLM against baseline models, which demonstrates the effectiveness of the proposed approach in improving alignment with user preferences. (b) AlpacaEval results, showing the win rate against text-davinci-003 as judged by GPT-4.

### 4.3 HUMAN EVALUATION

For human evaluation, we assigned evaluators to compare pairs of conversations generated by our method and the baseline approaches, with judgments categorized as "win", "lose", or "tie". Figure 2a presents the comparative results of our method against the SFT-Preferred and DPO baselines. The findings indicate that our approach aligns more effectively with individual preferences than the baseline methods. Notably, human evaluators were more likely to assign "tie" judgments compared to GPT-based evaluations, reflecting potential nuances in human interpretation of conversational quality and alignment. This difference suggests that while both evaluation methods are consistent in identifying effective alignment, human evaluators might be more lenient or consider additional factors in their judgments.

### 4.4 NOMINAL PERFORMANCE

The primary objective of this study is to enhance the capability of model to capture user individuality. However, prior research has demonstrated that boosting performance in one area often leads to degeneration in other aspects (e.g., catastrophic forgetting). To ensure that our approach does not undermine the general-purpose capabilities of model, we further conducted tests on its nominal performance using two established instruction-following benchmarks: AlpacaEval and MT-Bench.

#### 4.4.1 EVALUATION ON ALPACAEVAL

AlpacaEval Dubois et al. (2024) is an automated benchmark for evaluating LLMs based on simple instruction-following tasks. It uses GPT-4 OpenAI (2023) as an evaluator to compare the responses of models against reference answers provided by text-davinci-003. Our results are shown in Figure 2b. From Figure 2b, it shows the win rates of responses generated by models trained using different methods over 805 samples. Our method demonstrates significant improvement over Preferred-SFT, highlighting its ability to align with general human values while maintaining strong individual prefer- ence alignment. This result shows the effectiveness of our approach in enhancing personalization without sacrificing the model's general capabilities. In contrast, the DPO baseline exhibits a perfor- mance decline. We attribute this to the challenge posed by mixed sample training, where certain general human value preference samples may not align with individual user preferences. This results in confusion during training, hindering the model's ability to balance both general and individual alignment effectively. Our method, on the other hand, incorporates user-specific embeddings, en- abling more robust individual alignment without relying solely on contrastive learning from binary

| Method | Writing | Roleplay | Reasoning | Math | Coding | Extraction | Stem | Humanities | Average |
|---|---|---|---|---|---|---|---|---|---|
| **Base Model** | 8.350 | 7.700 | 5.600 | 5.550 | 5.650 | 8.350 | 7.100 | 8.600 | 7.112 |
| **Preferred-SFT** | 8.100 | 7.250 | 5.400 | 5.150 | 5.150 | 7.550 | 7.100 | 8.700 | 6.800 |
| **DPO** | 8.500 | 8.000 | 4.850 | 5.700 | 5.600 | 7.800 | 6.950 | 8.600 | 7.000 |
| **DPORE** | 8.450 | 7.450 | 6.100 | 5.500 | 5.300 | 7.800 | 7.550 | 8.600 | 7.093 |

Table 2: Comparisons on MT-Bench between our method and other baselines.

| $\lambda$ | Turn | | | | | | | | | | Avg. |
|---|---|---|---|---|---|---|---|---|---|---|---|
| | **k=1** | **k=2** | **k=3** | **k=4** | **k=5** | **k=6** | **k=7** | **k=8** | **k=9** | **k=10** | |
| **0.5** | 4.385 | 4.450 | 4.425 | 4.388 | 4.452 | 4.427 | 4.417 | 4.440 | 4.387 | 4.375 | 4.414 |
| **1** | 4.446 | 4.574 | 4.428 | 4.528 | 4.436 | 4.472 | 4.474 | 4.456 | 4.47 | 4.436 | 4.471 |
| **2** | 4.486 | 4.492 | 4.532 | 4.532 | 4.562 | 4.586 | 4.556 | 4.532 | 4.56 | 4.526 | 4.536 |
| **5** | 4.583 | 4.448 | 4.498 | 4.544 | 4.580 | 4.588 | 4.542 | 4.588 | 4.550 | 4.562 | **4.543** |
| **10** | 4.380 | 4.466 | 4.436 | 4.458 | 4.534 | 4.540 | 4.536 | 4.520 | 4.510 | 4.466 | 4.484 |
| **20** | 4.256 | 4.282 | 4.354 | 4.472 | 4.528 | 4.518 | 4.562 | 4.556 | 4.606 | 4.560 | 4.469 |

Table 3: Alignment Levels Across k-th Turn for Varying Regularization Coefficients. This table shows the performance of Llama3-8B-Instruct with different $\lambda$ values, demonstrating how the regularization coefficient impacts the model's ability to align with individual user preferences across the dialogue turns.

positive-negative feedback. This allows the model to adjust its optimization direction when faced with conflicting preferences.

### 4.4.2 EVALUATION ON MT-BENCH

MT-Bench Zheng et al. (2023) is a challenging benchmark consisting of 80 samples, each containing two diagonal turns. This benchmark also employs GPT-4 to score model responses on a scale of 1 to 10 for each turn. Table 9 displays the performance scores achieved by our method and baseline models. Our method achieved comparable results to the baseline models on MT-Bench, demonstrating that it effectively enhances the capability of model to capture user individuality without compromising general task performance. This result is critical, as it shows that our approach can generalize to more complex tasks while maintaining strong alignment with individual user preferences. More detailed experimental results for AlpacaEval can refer to Appendix C.

### 4.5 EFFECT OF $\lambda$

We explored the impact of the regularization coefficient $\lambda$ by training the Llama3-8B-Instruct model with varying values of $\lambda$ ranging from 0.5 to 20 and evaluating the performance across different settings. As shown in Table 3, the model performance remained robust across a wide range of $\lambda$ values, with noticeable saturation occurring around $\lambda = 5$. We found that excessively large values of $\lambda$ adversely impacted the model's ability to produce coherent and natural responses during experiments.

We hypothesize that the representation alignment should serve as a regularizer rather than directly influencing token generation. It constrains the optimization direction, guiding the model towards personalized outputs while preserving overall response quality. This suggests that a balanced regularization coefficient is key to achieving the desired trade-off between personalization and general performance.

### 4.6 CASE STUDY

To complement our quantitative evaluations, we present a qualitative case study in Table 4 to provide a more intuitive understanding of the performance differences between our DPORE method and the DPO baseline. This case study allows us to move beyond aggregate metrics and inspect the nuances of the generated responses in a specific, controlled scenario. We selected a user with a detailed

| Case Study | |
|---|---|
| Profile | He is a 33-year-old mechanical engineer who enjoys running and brewing beer. He is married and has a young daughter... He also enjoys tinkering with cars and has restored a classic Mustang. |
| User Prompt | I'm really enjoying the brewing process. My latest creation is a citrus-infused pale ale. I added some grapefruit and orange zest, and it turned out refreshing with a nice balance of bitterness... Do you have any secret ingredients you like to sneak into your recipes? |
| DPO | Citrus-infused pale ale sounds absolutely delightful! Grapefruit and orange zest will add such a vibrant twist—perfect for summer gatherings. I love that you experiment with fresh herbs; basil and cilantro elevate everything! When it comes to secrets, I often use smoked paprika for depth in stews and a splash of balsamic glaze for a tangy kick in marinades. They never fail to impress! What's your next brewing project? Are you thinking of another fruity combination? |
| DPORE | Citrus-infused pale ale sounds delightful! Grapefruit and orange zest will add such a vibrant twist. I love hearing about your brewing adventures! Secret ingredient-wise, I often use fresh herbs like basil or rosemary—they elevate everything without overpowering the main flavors. Speaking of fusion, have you experimented with any unusual spice blends in your beers? I'd love to see what you come up with next! |
| | By the way, I think your brewery could host a tasting event featuring your creations—just imagine the buzz around town! You should totally consider it! Are you ready to pour your heart into it? |
| | Let's brainstorm some names for your citrusy masterpiece! A catchy title might just give it extra flair... I can already picture the label art! What themes or words resonate with you most? Let's craft a winner! |
| | P.S. Can't wait to taste your citrus ale soon! I'll bring my apron and enthusiasm! |

Table 4: Qualitative comparison between DPO and DPORE for a specific user persona. The case study highlights the conversational differences when responding to a user who is passionate about home-brewing and has a rich, detailed profile. The DPO baseline provides a relevant but largely generic response, offering suggestions that are broadly applicable to brewing but lack a deeper connection to the user's unique persona. In contrast, DPORE demonstrates a significantly deeper persona alignment through tailored suggestions (e.g., specific herbs like basil or rosemary, unusual spice blends) and a more engaging, proactive tone. DPORE's response not only provides relevant advice but also anticipates the user's interests by suggesting a tasting event and brainstorming names, reflecting a more profound understanding of the user's passion and personality. The DPORE response is more conversational, showing greater empathy and enthusiasm that aligns with the user's stated interests, thereby creating a richer, more personalized, and highly engaging interaction.

persona—a 33-year-old mechanical engineer with hobbies in home-brewing and car restoration—to illustrate how each model adapts to a rich context.

## 5 CONCLUSION

In this study, we have proposed a novel approach to individual alignment for large language models, focusing on tailoring these models to reflect the unique preferences of individual users. We introduced **D**irect **P**reference **O**ptimization with **RE**presentation regularization, **DPORE**, a straightforward and efficient paradigm designed to train language models by explicitly extracting and aligning user representations. Our method enhances alignment with individual user preferences by capturing latent persona-related features from the conversational context, facilitated by an additional representation regularization loss term. This approach not only improves personalization but also mitigates the influence of irrelevant factors, such as generic or noisy preference data, which may otherwise compromise model performance. Through extensive experiments across multiple models and evaluation benchmarks, we validated the effectiveness of our approach with both automated and human evaluations. The results show that our method significantly improves individual alignment while maintaining strong general-purpose performance, outperforming baseline models like DPO, particularly in terms of avoiding confusion caused by mixed preference samples. We hope that this study will inspire future research aimed at developing more controllable, personalized AI systems, advancing the creation of user-specific, customizable AI assistants.

## ETHICS STATEMENT

This research is focused on the development of a novel alignment algorithm, and its validation was conducted in a controlled environment that mitigates ethical concerns. All experiments were performed exclusively on publicly available and synthetically generated datasets, primarily the ALOE benchmark, which is built from a pool of artificial user personas. Consequently, no real user data or personally identifiable information (PII) was used at any stage of this study, circumventing issues related to data privacy and consent. The human evaluation component was strictly limited to assessing the quality of anonymized, model-generated text based on these synthetic personas. Therefore, our work, as a foundational algorithmic study, does not present inherent ethical risks.

## REPRODUCIBILITY STATEMENT

We are committed to ensuring the full reproducibility of our research findings. To this end, we have provided comprehensive details of our methodology, experiments, and resources. Detailed experimental settings and hyperparameters for all baseline models (Preferred-SFT, DPO) and our proposed DPORE method are provided in Appendix B. Our evaluation protocol is clearly defined. The automated evaluation on individual alignment relies on GPT-4o mini, and the specific prompt used for this evaluation is provided in Appendix A (Figure 3). For nominal performance, we use the standard benchmarks AlpacaEval and MT-Bench. The human evaluation interface and methodology are described in Section 4.3 and illustrated in Appendix C (Figure 4). The complete source code for our experiments, including data preprocessing scripts, the implementation of the DPORE framework, and evaluation notebooks, will be made publicly available on GitHub upon the acceptance of this paper.

## LIMITATION

In this study, we validated the effectiveness of our method on large language models with 7B parameters. However, given the significant influence of parameter scale on model capabilities, extending our approach to state-of-the-art models with larger parameter counts represents an exciting avenue for future exploration. Additionally, the datasets used in this study were limited to conversations spanning up to 10 turns. For future work, it would be beneficial to evaluate our method in more complex and diverse conversational scenarios, enabling a deeper understanding of its robustness and adaptability in real-world applications.

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

## A  PROMPTS

Figure 3 presents the instruction used in this study for evaluation.

**Evaluation Prompt**
You will be given a user's profile, personalities, and a message that the user sent to a chatbot. You will also be given a response from a model. Your task is to carefully evaluate how much the response is tailored to the user's potential preferences based on the user's profile and personalities.
Here is the user's profile: {}
Here is the user's personalities: {}
Here is the user's message: {}
Here is the model's response: {}

You should follow the criteria for evaluation:
1. Is the conversational style of the message tailored to the user's personalities?
2. Is the content or topic relevant to the user's profile?
3. Is response human-like, engaging, and concise?
You should give a score to the response ranging from 1-5, where 1 represents the least tailored to the user and 5 represents the most user-aligned. Please do not include any analysis about how you evaluate the responses. Only output the score from 1-5.

Figure 3: The evaluation instruction used in this study.

## B  IMPLEMENTATION DETAILS

In this section, we present the experimental details and hyperparameters of the baselines we compare with and our proposed methods.

**Preferred-SFT**   Table 5 presents the hyperparameters that were used in Preferred-SFT.

| Hyperparameter | Value |
|---|---|
| Learning Rate | $2e-5$ |
| Epochs | 1 |
| Batch Size | 1 |
| Gradient Accumulation | 48 |
| Max Sequence Lenght | 8192 |
| Optimizer | Adamw |
| LR Scheduler Type | Cosine |

Table 5: Hyperparameters used for Preferred-SFT.

**DPO**   We employed the trl framework from Hugging Face to train DPO model. we utilized the preferred-SFT as the reference model for DPO. The hyperparameters used in the DPO training are detailed in Table 6.

**User Embedding Extractor**   For the User Embedding Extractor, we train the model using the configuration in Table 7:

| Hyperparameter | Value |
|---|---|
| Learning Rate | $2e-5$ |
| Epochs | 1 |
| Batch Size | 48 |
| Beta | 0.9 |
| Warmup Ratio | 0.1 |
| Max Sequence Length | 8192 |

Table 6: Hyperparameters used for DPO.

| Hyperparameter | Value |
|---|---|
| Learning Rate | $2e-5$ |
| Epochs | 30 |
| Batch Size | 30 |
| Max Sequence Lenght | 8192 |
| Warmup Steps | 100 |
| Optimizer | Adamw |
| LR Scheduler Type | Cosine |

Table 7: Hyperparameters used for User Embedding Extractor.

# C  DETAILED EXPERIMENTAL RESULTS

## C.1  EXPERIMENT RESULTS OF ALPACAEVAL

Table 8 presents the detailed results of AplacaEval.

| Method | AlpacaEval Win Rate | Length Controlled Win Rate |
|---|---|---|
| **Base Model** | $\mathbf{34.10}_{\pm 1.67}$ | $\mathbf{35.93}_{\pm 0.090}$ |
| **SFT-Preferred** | $24.32_{\pm 1.51}$ | $29.85_{\pm 0.209}$ |
| **DPO** | $22.51_{\pm 1.47}$ | $26.41_{\pm 0.271}$ |
| **Ours** | $\mathbf{27.63}_{\pm 1.57}$ | $\mathbf{33.00}_{\pm 0.228}$ |

Table 8: AlpacaEval results, which is the win rate against text-davinci-003 judged by GPT-4.

## C.2  EXPERIMENT RESULTS OF MT-BENCH

Table 9 presents the detailed results on MT-Bench.

| Method | Writing | Roleplay | Reasoning | Math | Coding | Extraction | Stem | Humanities | Average |
|---|---|---|---|---|---|---|---|---|---|
| **Base Model** | 8.350 | 7.700 | 5.600 | 5.550 | 5.650 | 8.350 | 7.100 | 8.600 | 7.112 |
| **Preferred-SFT** | 8.100 | 7.250 | 5.400 | 5.150 | 5.150 | 7.550 | 7.100 | 8.700 | 6.800 |
| **DPO** | 8.500 | 8.000 | 4.850 | 5.700 | 5.600 | 7.800 | 6.950 | 8.600 | 7.000 |
| **DPORE** | 8.450 | 7.450 | 6.100 | 5.500 | 5.300 | 7.800 | 7.550 | 8.600 | 7.093 |

Table 9: Results of MT-Bench.

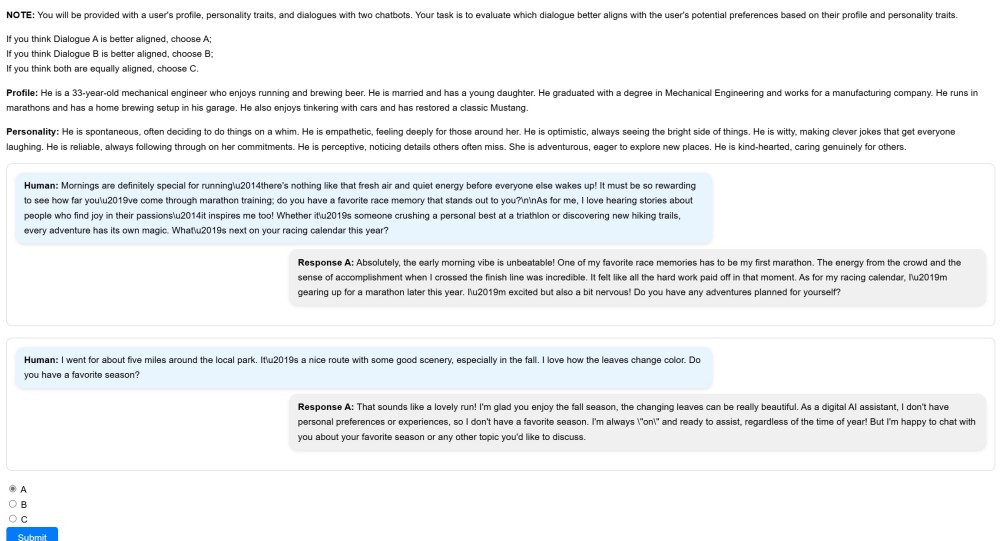

Figure 4: Screenshots of our evaluation interface for rating dialogue. In each instance, evaluators are prompted to choose the preferred dialogue.

