# OpenReview forum: "Context-Aware Alignment: Adapting Large Language Models to Individual Historical Data"
_ICLR.cc/2026/Conference — ICLR 2026 Conference Withdrawn Submission_

### Official Review · Reviewer_B1Pe · 2025-10-22

**Soundness:** 2
**Presentation:** 3
**Contribution:** 2
**Rating:** 2
**Confidence:** 4

**Summary:**

This paper introduces DPORE (Direct Preference Optimization with Representation Regularization), a novel approach for individual alignment of large language models (LLMs) that adapts models to user-specific preferences using historical conversational data. The core idea is to extract persona embeddings from past user interactions through a contrastive learning framework, ensuring embeddings capture abstract stylistic, tonal, and preference-related traits. These user representations are then integrated into DPO training via a representation regularization term, which encourages alignment between generated responses and user embeddings for preferred outputs while penalizing similarity for rejected ones. Experiments using the ALOE benchmark (3,000 multi-turn dialogues across 3,310 personas) demonstrate that DPORE significantly outperforms baselines (SFT, DPO) in individual alignment while maintaining comparable general-purpose performance on AlpacaEval and MT-Bench. A case study further shows that DPORE produces richer, persona-consistent dialogue compared to DPO.

**Strengths:**

1. This paper attempts to solve a relevant gap between general and individual alignment. Loss formulations and embedding pipeline are well defined.
2. The paper highlights the importance of integrating user context into preference learning.
3. Reproducibility details and open-source commitment are commendabl

**Weaknesses:**

1. All experiments rely on the ALOE benchmark, a fully synthetic dataset where both user personas and dialogues are generated by GPT models. While this setting allows controlled testing, it is not a realistic proxy for human personalization. Synthetic personas are typically coherent and idealized, lacking noise, ambiguity, or the inconsistencies that characterize real user histories. Thus, it is unclear whether DPORE would remain stable or meaningful on authentic human conversation logs, where linguistic variation and sparse feedback are dominant.
2. The contrastive embedding model is central to DPORE, but the paper provides no analysis of what these embeddings represent. There are no visualizations, clustering analyses, or qualitative examples demonstrating that they meaningfully separate users by personality, style, or topic preferences. Without interpretability studies, it remains unclear whether the embeddings encode genuine persona features or simply artifacts of text style.
3. The paper mentions human evaluation but provides no information about how it was conducted. How were evaluators recruited (e.g., expert annotators, crowd workers, internal team)? How many annotators participated, and were there any quality-control measures (inter-rater agreement, majority voting)? What evaluation questions or rubrics were used (e.g., were raters judging helpfulness, persona consistency, or both)? Without this information, it is impossible to assess the reliability, statistical validity, or replicability of the human results. This omission significantly weakens the empirical section.
4. GPT-4 is used to judge the outputs, but GPT-generated data were also used to train the synthetic personas and possibly influence the embedding extractor. This creates evaluation bias: models that align more closely with GPT’s writing style may appear better according to GPT’s own preferences. It's OK to use LLM as a judge, but it's not OK to have no human evaluation on human-LLM agreement.
5. The entire study uses only one model architecture (LLaMA-3 8B) for both training and evaluation. This makes it unclear whether the approach is model-agnostic or if the reported gains depend on a specific model’s embedding space and behavior. Testing across at least two different base models (e.g., Qwen, Mistral, Phi) would have made the claims more convincing.

**Questions:**

1. How exactly were human evaluators recruited, trained, and instructed? Please clarify the evaluation protocol and criteria.
2. How does DPORE perform when applied to a different base model?
3. Can the method generalize to new users with very limited history (few-shot scenarios)?

---

### Official Review · Reviewer_m543 · 2025-11-01

**Soundness:** 2
**Presentation:** 2
**Contribution:** 2
**Rating:** 4
**Confidence:** 3

**Summary:**

This paper addresses an important yet underexplored direction in LLM alignment: personalization to individual user preferences rather than only general human values. The authors propose a method that learns persona-aware representations from users’ historical interactions and uses them to guide response generation, achieving stronger individual alignment without compromising contextual coherence. Experiments on standard benchmarks show clear improvements over existing baselines, suggesting promising avenues for user-centric LLM deployment.

**Strengths:**

1. The proposed approach is technically sound and presents a valid strategy for personalized alignment by integration of persona-aware signals into the DPO framework
2. The paper is well-written, clearly structured, and easy to follow.

**Weaknesses:**

1.	The abstract is unclear and provides only a high-level, cursory description of the method—particularly regarding the extraction of persona-related representations. A more precise and informative summary would better convey the technical contribution.
2.	Evaluation is limited to the ALOE benchmark with only 100 samples, which is insufficient to assess the generalizability of the approach. The authors should include additional personalized alignment benchmarks (e.g., LongLaMP [1], PERSONA [2], PREFEVAL [3], or others) to demonstrate robustness across diverse settings.
3.	Key implementation details are missing: the architecture and hyperparameters of the encoder G are not specified, and equation numbering appears to be inconsistent (e.g., equations after Eq. 1 are unnumbered).
4.	The core idea—injecting user-specific representations into DPO via a regularizer—is relatively straightforward and closely follows existing alignment frameworks. Given the limited experiments, the novelty and contribution appear modest.

**Questions:**

See the weaknesses.

---

### Official Review · Reviewer_kBmr · 2025-11-02

**Soundness:** 2
**Presentation:** 3
**Contribution:** 2
**Rating:** 2
**Confidence:** 4

**Summary:**

This paper introduces DPORE, a method for context-aware personalization of large language models that extends DPO by incorporating persona embeddings learned from historical conversations. These embeddings serve as stylistic and preference representations that guide alignment through a representation-regularized loss. The paper presents a clear motivation and shows moderate improvements on the ALOE benchmark with a narrow evaluation. However, the paper lacks critical baselines and generalization tests, which weakens the paper’s empirical foundation. While the idea of embedding-based personalization is promising, the evidence does not convincingly demonstrate that DPORE provides substantial or generalizable gains over simpler alternatives. As such, the broader claims of effective and generalizable personalization are not substantiated.

**Strengths:**

- Addresses an important and emerging problem: scalable, individualized adaptation of LLMs.
- Methodologically sound extension of DPO with representation regularization.
- Results suggest some improvements wrt persona consistency and stylistic coherence in multi-turn dialogue.

**Weaknesses:**

- Limited evaluation. The model is tested on a single synthetic dataset with only 100 evaluation cases, offering a weak empirical basis for general conclusions about personalization performance.
- Missing baselines. The paper omits an in-context personalization baseline (e.g., adding user profile information to the prompt) which is essential for judging whether fine-tuning offers meaningful improvement beyond prompting.
- Lack of generalization analysis. The paper does not examine how a persona-tuned model generalizes to unseen personas or to users without profiles, leaving unclear whether DPORE overfits to training identities or degrades in default use.

**Questions:**

See weaknesses.

---

### Official Review · Reviewer_G9oY · 2025-11-08

**Soundness:** 2
**Presentation:** 4
**Contribution:** 2
**Rating:** 2
**Confidence:** 3

**Summary:**

This paper addresses individual alignment, where user-specific preferences may differ from the general preferences learned through RLHF. To achieve this, the authors propose a user embedding learning step using a contrastive learning objective, encouraging dialogue excerpts from the same persona to be closer in representation space and farther apart otherwise. They further incorporate a persona-based regularization term into the DPO objective, based on the similarity between response representation and the learnt user embeddings. Experiments are conducted on the ALOE benchmark, comparing against base models, SFT on preferred responses, and vanilla DPO.

**Strengths:**

The paper tackles an important and underexplored problem—aligning LLMs to individual rather than collective preferences, which is a known limitation of current alignment frameworks such as DPO.

The proposed method of learning user embeddings and integrating them as a regularization term is conceptually simple, intuitive, and shows improved results over vanilla DPO without degrading general language understanding or reasoning performance.

The paper is clearly written and easy to follow.

**Weaknesses:**

The proposed persona-based regularization lacks theoretical grounding. There is no analysis explaining why this specific regularization form benefits DPO the most or how it affects its optimality. The design choices are intuitive but unsupported by deeper reasoning or ablations.

The method assumes user identity/persona labels are available for all dialogue excerpts to train the contrastive objective. In real-world scenarios, such annotations are rarely accessible. It is also unclear whether the learned persona embeddings generalize to unseen users at inference time, as no cross-user generalization experiments are provided.

Both the contrastive learning and evaluation are performed on the ALOE dataset, with overlapping persona pools. This raises a concern about data leakage. If the same personas appear in both stages, the model’s apparent ability to “recover” personas may simply reflect memorization.

The experiments only include basic ablations (Base, SFT, and vanilla DPO). The authors do not follow the full evaluation setup of ALOE, omitting alignment-level metrics and normalized IR scores. The regularization should, in principle, be applicable to other RLHF methods (e.g., PPO), yet no such experiments are presented to demonstrate generality.

The performance gain on AlpacaEval requires more explanation. Case studies suggest that DPORE tends to generate longer responses to appear more “personal,” which could inflate scores due to length bias rather than genuine personalization.

**Questions:**

What data is used for DPO training in DPORE? Are the training data and persona pools in the contrastive learning stage the same as those used in evaluation?

Could you provide the original evaluation metrics from ALOE for comparison?

---

### Note · Authors · 2025-11-12

**Comment:**

Dear Reviewers,

We sincerely appreciate the time and effort you have dedicated to reviewing our submission. Your valuable feedback has provided us with significant insights, which will be invaluable as we continue to refine our work.

After careful consideration, we have decided to withdraw our paper from the conference. We believe this decision will allow us to further improve the quality of our research before presenting it to the community.

Thank you once again for your understanding and support.

**Withdrawal Confirmation:**

I have read and agree with the venue's withdrawal policy on behalf of myself and my co-authors.